# The efficiency in the ordinary hospital bed management: A comparative analysis in four European countries before the COVID-19 outbreak

**Fabrizio Pecoraro**[1]*, **Daniela Luzi**[1], **Fabrizio Clemente**[2]

1 Institute for Research on Population and Social Policies, National Research Council, Rome, Italy,
2 Institute of Crystallography, National Research Council, Monterotondo, Rome, Italy

* f.pecoraro@irpps.cnr.it

**Data Availability Statement:** Raw data has been published to Zenodo repository and is available at the following link: 10.5281/zenodo.3737839.

## Abstract

During COVID-19 emergency the majority of health structures in Europe saturated or nearly saturated their availabilities already in the first weeks of the epidemic period especially in some regions of Italy and Spain. The aim of this study is to analyse the efficiency in the management of hospital beds before the COVID-19 outbreak at regional level in France, Germany, Italy and Spain. This analysis can indicate a reference point for future analysis on resource management in emergency periods and help hospital managers, emergency planners as well as policy makers to put in place a rapid and effective response to an emergency situation. The results of this study clearly underline that France and Germany could rely on the robust structural components of the hospital system, compared to Italy and Spain. Presumably, this might have had an impact on the efficacy in the management of the COVID-19 diffusion. In particular, the high availability of beds in the majority of the France regions paired with the low occupancy rate and high turnover interval led these regions to have a high number of available beds. Consider also that this country generally manages complex cases. A similar structural component is present in the German regions where the number of available beds is significantly higher than in the other countries. The impact of the COVID-19 was completely different in Italy and Spain that had to deal with a relevant large number of patients relying on a reduced number of both hospital beds and professionals. A further critical factor compared to France and Germany concerns the dissimilar distribution of cases across regions. Even if in these countries the hospital beds were efficiently managed, the concentration of hospitalized patients and the scarcity of beds have put pressure on the hospital systems.

## Introduction

The severe acute respiratory syndrome coronavirus 2 (SARS-CoV-2), the virus causing coronavirus disease 2019 (COVID-19), has dramatically infected a significant number of

**Funding:** The authors received no specific funding for this work.

**Competing interests:** The authors have declared that no competing interests exist.

individuals worldwide [1]. The first person-to-person transmissions in Europe were reported at the end of February 2020, and led to an infection chain that represents one of the largest outbreaks occurred in the last centuries. From these first cases a rapidly increasing number of patients have been identified, initially in Northern Italy and later in the rest of the country, and subsequently in Spain, France, Germany [1–3]. At the moment (i.e. October 8th, 2020), after a relative deceleration during the summer period, the virus has continued to rise in whole Europe reaching 5.644.987 confirmed cases (15% of all cases worldwide) and causing 228.053 deaths (21% of all deaths worldwide) [1, 4]. These numbers resulted in a critical access to healthcare facilities in particular during the first period of the epidemic outbreak making it difficult to treat not only patients clearly affected by the COVID-19 (i.e. with serious symptoms), but also individuals with light symptoms referable to the virus as well as to other common pathologies. Unless the epidemic curve flattens over a long period, the high consumption of healthcare resources has likely to cause a shortage of both Intensive Care Unit (ICU) and ward hospital beds as well as medical equipment such as ventilators. Moreover, the treatment of patients affected by COVID-19 may have impacted on healthcare workers also considering that doctors, nurses and other professionals may became ill or quarantined especially in those countries where the majority of patients were hospitalized instead of confined at home [5, 6]. As reported by the Italian Ministry of Health [7–9], during the first weeks of the epidemic spread, about 20–30% of cases of COVID-19 in Europe were hospitalised with a higher rate of people over 60 years of age and comorbidities and 4% of them presented a severe status due to COVID-19 infection disease. This resulted in an overuse of hospital beds that led researchers as well as journalists to criticize the availability of such resources, in particular in Italy [10–12] highlighting that the past reductions in the number of beds have affected many hospitals over the territory. This reduction, which is not limited only to Italy, is mainly related to the progressive cutting of health expenditure at the national and local level which resulted in an unbalanced number of beds over the population [13]. Similar results have been published in the literature considering the number of beds in the ICU departments [14].

Starting from these premises, this study analyses the regional hospital structural components before the outbreak of the COVID-19 in four European Countries: France, Germany, Italy and Spain. The paper aims to assess the efficiency and the performance of the hospital bed management in the past years, that in our view can indicate a reference point for future analysis on resource management in emergency periods. It can help hospital managers, emergency planners as well as policy makers to put in place a rapid and effective response to emergency such as the COVID-19 outbreak. As reported by the WHO [15] one of the crucial actions to ensure a rapid response to the COVID-19 outbreak is the ability of a health service to expand beyond its normal capacity to meet an increased demand for clinical care. In this perspective the availability of hospital beds as well as the efficiency in the management of the health resources play a crucial role to determine the room for manoeuvre of the healthcare facilities in case of an emergency situation, such as the COVID-19. Moreover, this study may help to determine, whether the recent reduction of hospital resources may have had an impact on the functioning of hospitals and on the efficiency in the management of clinical cases [16]. In this perspective, two methodologies have been adopted to assess the efficiency of a health structure in the management of clinical hospitalized cases. The first one is the hospital bed management [17–19] that provides an overall description of the use of beds by health structures. The second methodology evaluates the performance of a hospital considering the complexity of cases treated by the structure [20, 21]. Both methodologies investigate hospital performances providing a helpful snapshot for healthcare managers for the evaluation of healthcare systems [22]. The paper is structured as follows: after the materials and methods paragraph, a

comparison of data captured in the years 2012 and 2017 is performed to verify differences across years in terms of hospital resources available in each region. Next, an overview of the virus diffusion and workload of the hospital infrastructure is reported to provide an overview of the extraordinary event faced by the health and social care professionals in the analysed countries. Finally, the results of the hospital bed management analysis are reported to provide an overall assessment of the efficiency as well as the complexity and performance of patient hospitalizations in the analysed countries.

## Materials and methods

### Data

This analysis explores the efficiency in the management of hospital beds focusing the attention on four European countries: France, Germany, Italy and Spain. As previously reported, we selected these countries as they were the first affected by the coronavirus and still remain those with the highest number of infections and deaths in Europe [1, 2, 3, 4]. In order to analyse potential regional differences within each country, the following regions were included in the study:

1)  Germany and Italy: all regions were included in the analysis

2)  Spain: all European continental regions as well as Canary and Baleares Islands (autonomous cities of Ceuta and Melilla were excluded from the analysis)

3)  France: all European continental regions as well as Corse Island (La Reunion, Martinique, Guadeloupe, Guyana and Mayotte were excluded from the analysis).

Data needed to compute the hospital bed management indicators were captured from the Eurostat database [23] distributed by region (NUTS 2: Nomenclature for Territorial Units for Statistics–basic regions for the application of regional policies). In particular:

- Hospital beds by NUTS 2 regions online (i.e. Eurostat data code: HLTH_RS_BDSRG);

- Hospital discharges by diagnosis (i.e. ICD-10) and NUTS 2 regions, in-patients, total number—total (i.e. Eurostat online data code: HLTH_CO_DISCH1T);

- In-patient average length of stay (days) by diagnosis (i.e. ICD-10) and NUTS 2 regions–total (i.e. Eurostat online data code: HLTH_CO_INPSTT).

Additional data captured from the Eurostat database [23] were included in the study to analyse the structural components of each country. In particular:

- Health care expenditure (% Gross Domestic Product, GDP) by function online (i.e. Eurostat data code: HLTH_SHA11_HC) taking into account the total amount as well as the resources invested in the in-patient, outpatient and home care;

- Physicians by medical speciality (i.e. Eurostat data code: HLTH_RS_SPEC) and Health personnel (i.e. Eurostat data code: HLTH_RS_PRSRG) to capture the number of primary care physicians, specialists in general as well as hospital medical doctors and hospital nurses and midwifes per 100.000 inhabitants.

Each analysis compared the last available data (i.e. 2017) with those related to 2012 to capture differences in terms of investment and resources in each country as well as possible significant changes across 5 years. These analyses were performed at national level on the basis of data availability.

## Hospital bed management

The overall description of the hospital bed management was assessed using the following indicators computed on the basis of the number of beds, the patient discharged over a specific period of time (i.e. a year) and total number of in-patient days (i.e. the overall number of days that all the patients are hospitalized) [24].

- Beds Occupancy Rate (BOR): percentage of inpatient beds occupied over a specific period;

- Average Length Of Stay (AvLOS): average number of days that an inpatient remained in the hospital;

- Turnover Interval (TOI): number of days that an available bed remains empty between the discharge of a patient and the admission of a next one;

- Beds Turn Over (BTO): average number of patients "passing through" each bed during a specific period.

These values were presented adopting the Barber-Johnson diagram [25] that allows to combine in a unique scatterplot diagram the four-mentioned variables. For the target indices thresholds values for TOI (1<TOI<3) and BOR (75%<BOR<85%) [26–29] have been adopted. This diagram provides a prompt graphical view of the efficiency in the management of hospital beds classifying each region in the following areas: 1) the red one that identifies regions where both TOI and BOR are outside the reference threshold; 2) the yellow area that reports regions where either TOI or BOR are outside the threshold and 3) the green area that identifies regions where both indicators are within the reference thresholds.

The complexity and the performance of each clinical department in the management of clinical hospitalization cases were described, respectively, by the Case Mix (CMI) and the Performance Index (PI). In particular, the former indicates the degree of complexity in each observed unit with respect to a benchmark level (i.e. the national average), while the PI compares the performance of the observed unit with the benchmark level on the basis of the inpatient length of stay. These indicators were computed according to the following formulas, taking into account the AvLOS of the different diseases addressed by the relevant structure.

$$CMI = \frac{\sum_{j=1}^{n} AvLOS_j^{REF}\left(\frac{d_j^{STR}}{d_{ALL}^{STR}}\right)}{AvLOS_{ALL}^{REF}} \tag{1}$$

$$PI = \frac{\sum_{j=1}^{n} AvLOS_j^{STR}\left(\frac{d_j^{REF}}{d_{ALL}^{REF}}\right)}{AvLOS_{ALL}^{REF}} \tag{2}$$

Where the variables indicate:

- $n$: number of specialties assessed by the structure;

- $AvLOS_j^{REF}$: reference value of the AvLOS of the specialty $j$;

- $AvLOS_{ALL}^{REF}$: reference value of the AvLOS of all specialties;

- $d_j^{REF}$: total number of patient discharges in the four countries of the specialty $j$;

- $d_{ALL}^{REF}$: total number of patient discharges in the four countries;

- $AvLOS_j^{STR}$: AvLOS of the specialty $j$ in the relevant structure;

- $d_j^{STR}$: number of patients discharged in the relevant structure for the specialty j;

- $d_{ALL}^{STR}$: total number of patients discharges in the relevant structure.

A high CMI value (Eq 1) indicates a more complex and resource-intensive case load managing of clinical cases. In particular, a region with a CMI higher than 1 tends to hospitalize patients in wards with a high average length of stay in comparison to the benchmark (i.e. note that the benchmark is computed considering the four countries involved in the present study). Conversely, a high PI (Eq 2) value is found when the hospital length of stay is longer than expected. In particular, a region with a PI higher than 1, assuming equal complexity, tends to hospitalize patients for longer periods, thus suggesting lower efficiency relative to the standard [20, 21, 30]. Considering the graphical representation, these indicators were analysed adopting a four-quadrant graph where the CMI is reported in the abscissa and the PI is reported in the ordinate compared to the national benchmark. This representation allows to identify four specific areas: 1) a red one that identifies regions where both CMI and PI values are outside the relevant thresholds. This area identifies regions that manage low complexity cases with low performance; 2) a yellow zone when the CMI is outside the relevant threshold while the PI reaches the target. It identifies regions that efficiently manage low complexity cases; 3) an orange zone when the PI is outside the given threshold. In this area regions manage high complexity cases but with low performances; 4) a green zone where both indicators are within the given thresholds. In this area regions efficiently manage high complexity cases.

Both analyses were performed at regional level to capture differences between and within countries. Moreover, the results of these two analyses were combined to show the overall behaviour of each region in the management of hospital beds. The main clusters are summarized in Fig 1 highlighting a joint reading of the efficiency in the hospital beds management (x-axis) and the performance and complexity of cases managed in the hospital (y-axis).

## Results

### Structural components

**Health expenditure.** Fig 2 reports the trend over the past years of the health expenditure (public and private) in the four analysed countries. As reported by Eurostat this indicator quantifies the economic resources dedicated to health functions specifically concerning healthcare goods and services that are consumed by resident units. Germany is the only country that shows a continuous increase in the health expenditure (from 10,8 in 2012 to 11,3 in 2017), while the other three countries maintain the expenditure unchanged, with a slight reduction in Italy and Spain. Differences between northern countries (i.e. Germany, France) and southern countries (i.e. Italy, Spain) are evident.

The differences of expenditure (2012–2017) for inpatient, outpatient and home care patients are reported (Table 1). Italy is the only country that highly reduced the investment of inpatient care, increasing however health expenditures in the outpatient care. In addition, in this case differences between northern countries (i.e. Germany, France) and southern countries (i.e. Italy, Spain) are notable in particular considering the inpatient care.

**Health personnel.** Remarkable differences between countries can be detected considering the healthcare personnel per population (Table 2), in particular hospital medical doctors and hospital nurses and midwifes. As for the health expenditure, Italy and Spain show the lower number of hospital personnel, in comparison with France and Germany. Moreover, Italy is the country with the highest reduction of personnel during the last five years compared to the other countries. The slight increase of specialists, shown especially in Germany, may indicate the tendency to privilege specialized medical branches.

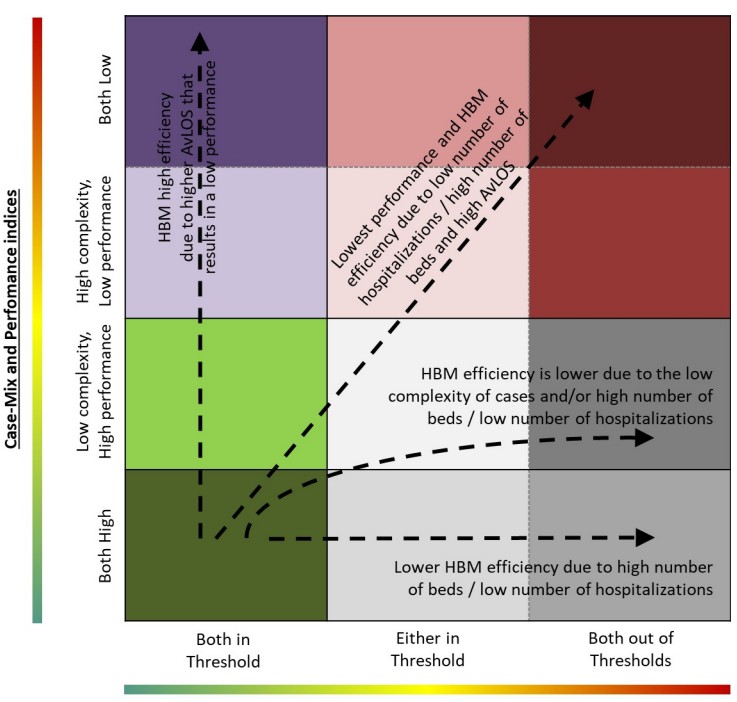

**Fig 1. Overall clustering considering the hospital bed management (x-axis) and the case-mix and performance indicators (y-axis).**

**Hospital beds.** The map and the histogram shown in Fig 3 highlight the regions with a high number of beds per population. As clearly reported, Germany and a limited number of regions in France have the highest number of beds per 100.000 inhabitants (reported in green), while Spain and Italy have the lowest values (reported in dark red). A clear difference is highlighted comparing the number of beds available in Germany where the region with the lowest value (Berlin, N = 580 per 100.000 inhabitants) counts more beds than the region with the highest values in Spain (Catalonia, N = 388) and Italy (Emilia Romagna, N = 378). This is showed in the histogram that reports, for each country, the region with the minimum and the maximum number of beds per 100.000 inhabitants (e.g. Germany spans from 580 to 1285, Italy from 250 to 387, Spain from 219 to 388 and France from 517 to 819). The histogram also

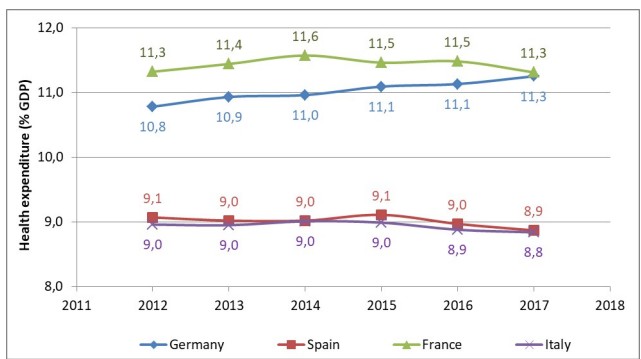

**Fig 2. Health expenditure (public and private) reported as percentage of the GDP.**

**Table 1. Health expenditure (public and private) reported as percentage of the GDP in 2017 and in percentage variation with 2012.**

| Expenditure % GDP | Inpatient care | | Outpatient care | | Home care | | Total | |
|---|---|---|---|---|---|---|---|---|
| | 2017 | % diff 2017–2012 | 2017 | % diff 2017–2012 | 2017 | % diff 2017–2012 | 2017 | % diff 2017–2012 |
| France | 3,12 | -2% | 2,05 | 11,31 | 0% | 2% | 0,42 | 14% |
| Germany | 2,98 | 0% | 2,44 | 11,25 | 4% | 0% | 0,07 | 0% |
| Italy | 2,45 | -9% | 2,03 | 8,84 | -1% | 7% | 0,03 | 0% |
| Spain | 2,16 | 0% | 2,77 | 8,87 | -2% | -5% | 0,07 | -13% |

highlights intra-country variability with the lowest differences across regions in Italy and Spain and the highest differences in Germany.

Compared with bed availability in 2012 (Fig 4), only 8 regions, all located in Spain, have increased the number of beds in the period 2012–2017. This increase in hospital beds has affected in particular the Spanish regions with the lowest values of hospital bed per population in 2012, probably indicating interventions aiming at the reduction of regional disparities in health resources. As a result, Spain is the country with the lowest reduction of beds (around 1%, going from 299 beds per 100.000 inhabitants in 2012 to 297 in 2012) in comparison with Germany (-4%, from 834 to 800), France (-6%, from 633 to 598) and Italy (-7%, from 342 to 318). From a regional perspective, differences span from Sardegna (Italy, -13%) to La Rioja (Spain, +13%) as highlighted in the map.

## Impact of the COVID-19

This paragraph provides an overview of the diffusion of the COVID-19 across the four countries investigated in this paper considering data collected until 14[th] of July 2020, when the number of cases and deaths as well as the access to hospital facilities have reached a substantial reduction in these countries. Data are retrieved at a country and regional level from the following institutional web sites:

- Germany: Robert Koch Institute [31].

- France: Open platform for French public data [32].

- Italy: Civil Protection Department [33].

- Spain: Ministry of Health–Coordination Centre for Health Alerts and Emergencies (CCAES) [34].

As data are heterogeneously collected across countries, a comparison at regional level was difficult. This is evident considering data concerning the hospitalization rate across each country. For instance, while Italy, France and Spain collect the daily number of patients hospitalized, Germany provides this information only at a national level considering the cumulative

**Table 2. Health personnel per 100.000 inhabitants in 2017 and variation with 2012.**

| | General Practitioners | | Paediatrician | | Specialist | | Hospital Medical Doctor | | Hospital nurses and midwifes | |
|---|---|---|---|---|---|---|---|---|---|---|
| | 2017 | % diff 2017–2012 | 2017 | % diff 2017–2012 | 2017 | % diff 2017–2012 | 2017 | % diff 2017–2012 | 2017 | % diff 2017–2012 |
| France | 90,05 | -7% | 12,08 | 6% | 76,15 | 6% | 260,98 | 3% | 572,74 | 0% |
| Germany | 70,38 | 0% | 16,83 | 4% | 134,79 | 10% | 237,12 | 13% | 560,15 | 7% |
| Italy | 72,24 | -5% | 28,67 | -1% | 143,97 | 5% | 215,04 | 0% | 432,02 | -4% |
| Spain | 75,93 | 1% | 26,59 | 3% | 106,52 | 6% | 231,33 | 5% | 341,50 | 9% |

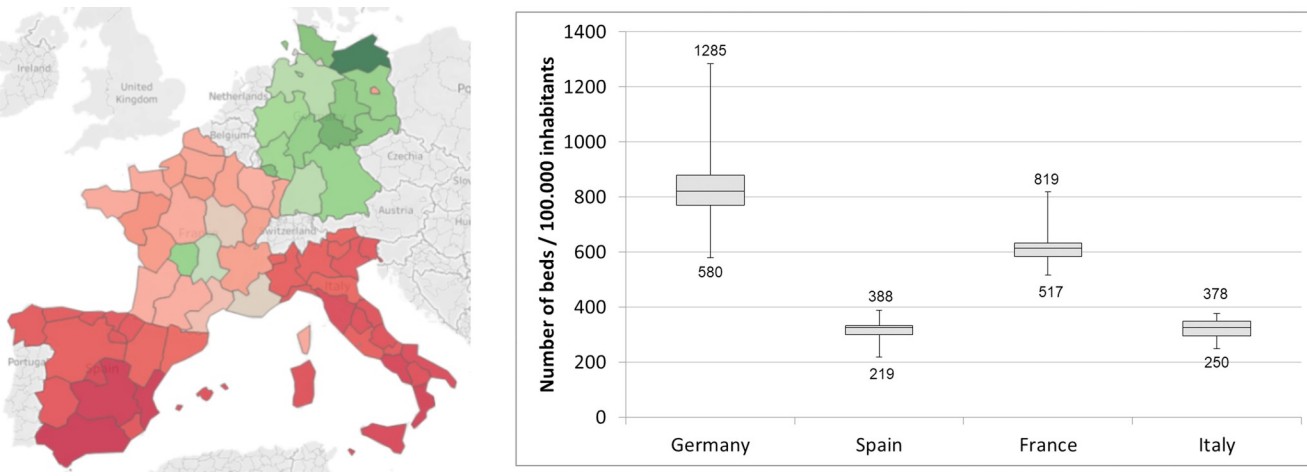

**Fig 3. Number of hospital beds per 100.000 inhabitants: (a) Distribution by region (map) where the colours span from red to green to specify respectively regions with the highest and the lowest number of beds per 100.000 inhabitants (map was created on the basis of Eurostat data [23] using Tableau Software; Seattle, Washington, United States); (b) Numerical distribution for each nation.**

number of patients. Moreover, the number of cases across each country are collected differently: France provides only the national cumulative aggregation, whereas the other three countries report data at a regional level. To fill this methodological gap, data provided by the above institutional web sites were integrated with those reported by the Institute for Health Metrics and Evaluation (IHME) [35] that forecasts the access to hospital services, daily and cumulative deaths due to COVID-19, rates of infection and testing, organized by country and regions/states.

What is clear from this preliminary analysis is that since the beginning of the contagion, the number of patients infected by COVID-19 are not homogenously distributed both between and within the four analysed countries. Table 3 summarizes the main indices of the COVID-19 diffusion taking into account the number of patients infected, the mortality as well as the number and percentage of patients hospitalized. As previously reported and highlighted in the table notes, some of these results have been not homogenously computed and cannot be directly compared. However, this table intends to highlight at a high level the main differences between countries to better contextualize the results reported in the next paragraphs.

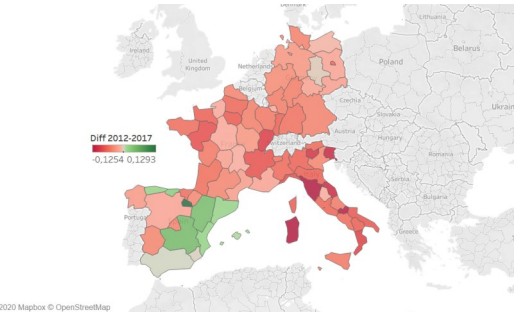

**Fig 4. Variation (2012–2017) of the number of hospital beds per 100.000 inhabitants.** Each region is coloured on the basis of the difference between the number of beds in 2012 and in 2017: dark red reported the highest reduction (i.e. Sardegna), dark green reported the highest increase (i.e. La Rioja) (map was created on the basis of Eurostat data [23] using Tableau Software; Seattle, Washington, United States).

**Table 3. Diffusion of the COVID-19 across the four analysed countries (as of 14th of July 2020) considering: Cases, death and hospitalizations in wards and intensive care units.**

|  |  | France | Germany | Italy | Spain |
|---|---|---|---|---|---|
| Cases | Number | 266,8 | 240,2 | 402,2 | 547,6 |
|  | CoV | 0,39* | 0,38 | 0,77 | 0,68 |
| Deaths | Number | 30,0 | 10,9 | 57,8 | 60,4 |
|  | CoV | 0,64 | 0,49 | 0,76 | 0,79 |
| Ward admission | % cases | 40% | 17% | 23% | 49% |
|  | Number | 105,4** | 40,8** | 91,1** | 220,7** |
|  | CoV | 0,79 | 0,79 | 0,95 | 0,81 |
| ICU admission | % cases | 21% | 8% | 3% | 5% |
|  | Number | 56,8** | 19,2** | 0,40** | 19,2** |
|  | CoV | 0,79 | 0,79 | 0,83 | 0,67 |

Notes:

* In France the number of cases at regional level are provided from the 18th of May.

** In Germany hospitalization data at regional level are not reported. Data in the table is computed considering the estimation reported in [35]. In Italy and Spain the number of patients hospitalized are computed considering the average number of patients admitted in the relevant period. In France ward and ICU admissions are jointly reported. The subdivision between ward and ICU admissions is computed considering the estimation reported in [35].

Data are reported as per 100.000 inhabitants. The CoV shows the variability across regions.

Moreover, the adoption of the Coefficient of Variation (CoV) makes it possible to underline how the virus is differently distributed even at a country level. The number of cases as well as the mortality rate clearly point out that Germany and France (even if the latter with less evidence) have been less affected by the virus than Spain and Italy. This is particularly evident in Germany where the number of deaths per 100.000 inhabitants were 10,9 in contrast to the data reported in France (N = 30,0), Italy (N = 57,8) and Spain (N = 60,4). Similarly, Germany counted a limited number of patients affected by the virus (N = 240,2) compared with France (N = 266,8), Italy (N = 402,2) and Spain (N = 547,6). Moreover, considering the CoV Germany showed the lowest rates implying that in this country the number of cases as well as the number of deaths are homogenously distributed across regions. A similar result is present in France but for the distribution of cases it is important to note that data at regional level are provided from the 18th of May. On the contrary, Italy and Spain showed the highest rates implying a critical management of healthcare resources in specific regions/areas of the country.

Considering the use of the hospital resources, even if Germany counts a high availability of hospital beds per 100000 inhabitants, it is the country with the lowest number of hospitalizations in the general wards (N = 40,8) compared with the other countries (Italy, N = 91,1; France, N = 105,4 and Spain, N = 220,7). Differently, the access to ICU beds was significantly higher in France (N = 56,8 equal to 21% of the total patients affected by COVID-19) when compared to Italy (N = 0,40; 3%), Spain (N = 19,2; 5%) and Germany (N = 19,2; 8%). In Italy the relative limited number of hospitalizations during the period both in the general wards and in the ICU is to be counterbalanced with the high variability of the disease diffusion that overwhelmed the capacity of hospitals and the health system at large in particular in the northern and central part of the country.

Even if the lack of hospitalization data as well as the continuous change in terms of bed availability across the European regions make it difficult to apply this methodology in this pandemic period, data presented in this paragraph clearly highlights the critical role played by hospitals within the health system in providing essential medical care to the community. Moreover, even if this analysis has been performed only at an aggregate national level, it is

important to note that the virus has had a different impact not only among the four countries analysed, but also at a regional level in particular in Italy and Spain. For this reason, the next paragraph intends to investigate whether regional differences in the efficiency in the management of hospital beds may have had an impact in the diffusion and mortality of COVID-19. Of course, this study provides only an overall ex-ante indication that might be further explored in a future analysis which should rely on updated and robust data so to provide indications on the hospital-territory relationship in response to citizens' safety and wellbeing.

## Hospital bed management

Fig 5A depicts the Barber-Johnson diagram that promptly classifies each region in the following coloured areas taking into account the efficiency in the management of hospital beds (see S1 Table for detailed data): 1) the red one identifies regions where both TOI and BOR are outside the reference threshold; 2) the yellow area reports regions where either TOI or BOR are outside the threshold and 3) the green area identifies regions where both indicators are within the reference thresholds indicating an efficient management of bed turnover and occupancy. Distribution of regions within these areas are shown in the histogram (Fig 5B) as well as in the map (Fig 5C):

1) Italy is the country with the highest number of regions within the green area (76%, 16 out of 21), followed by Germany (69%, N = 11). Moreover, both countries have the lowest number of regions within the yellow (10%, N = 2) and red (14%, N = 3) areas.

2) France and Spain are the countries with the highest number of regions within the red area (France: 50%, N = 11; Spain: 58%, N = 11). Spain also has a significant number of regions within the yellow area (26%, N = 5).

## Complexity and performance analysis

The diagram reported in Fig 6A highlights the classification of each region within four coloured zones on the basis of the inpatient average length of stay (see S1 Table for detailed data): 1) red zones when both CMI and PI values are outside the relevant thresholds. This area identifies regions that manage low complexity cases with low performance; 2) yellow zone when either the CMI is outside the relevant threshold while the PI reaches the target. It identifies regions that efficiently manage low complexity cases; 3) orange zone when the PI is outside the given threshold. In this area regions manage high complexity cases, but with low performances; 4) green zone where both indicators are within the given thresholds. In this area regions efficiently manage high complexity cases. Distribution of regions within these areas are shown in the histogram (Fig 6B) and in the map (Fig 6C). Note that the map colour span from green to red depending on the CMI and PI values.

1) France is the only country with all regions within the green area. The remaining regions within this area are located in Germany (25%, N = 4). Moreover, Germany has no regions within the red area, while the remaining 12 regions are equally distributed between the yellow and the orange area (Fig 6A);

2) All the regions in Spain and Italy manage low complexity cases. In particular, Italy places the majority of its region within the yellow area (86%, N = 18) confirming a high performance in the management of low complex cases and the remaining three (14%) in the red area. A similar result can be found in Spain with 11 regions (58%) in the yellow area and 8 (42%) in the red area. This result is also shown in the map (Fig 6C) highlighting that red regions (low complexity and low performance) are mainly distributed in the north side of Spain, Italy and Germany.

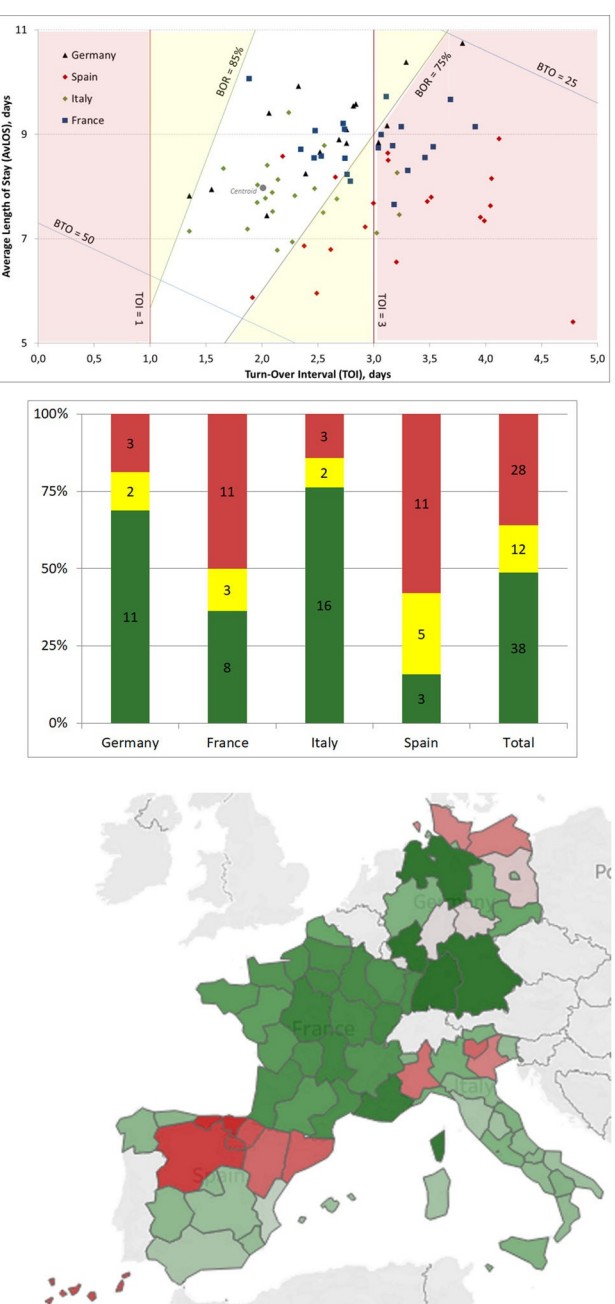

**Fig 5. Efficiency results of countries' regions in the management of hospital beds: (a) Barber-Johnson diagram highlighting the three areas and the centroid (TOI = 2 and BOR = 80%); (b) distribution of regions by country and by area and (c) map of the regions.** Each region is coloured on the basis of the Euclidean distance from the centroid: dark red reported the highest distance, dark green reported the lowest distance (map was created on the basis of Eurostat data [23] using Tableau Software; Seattle, Washington, United States).

## Overall analysis

Results reported in the previous paragraph are summarized in Fig 7 synthetizing the classification of each region in the clusters identified in the materials and methods section (see Fig 1). As said, clusters reported in Fig 7 combine the results of the hospital bed management (x-axis)

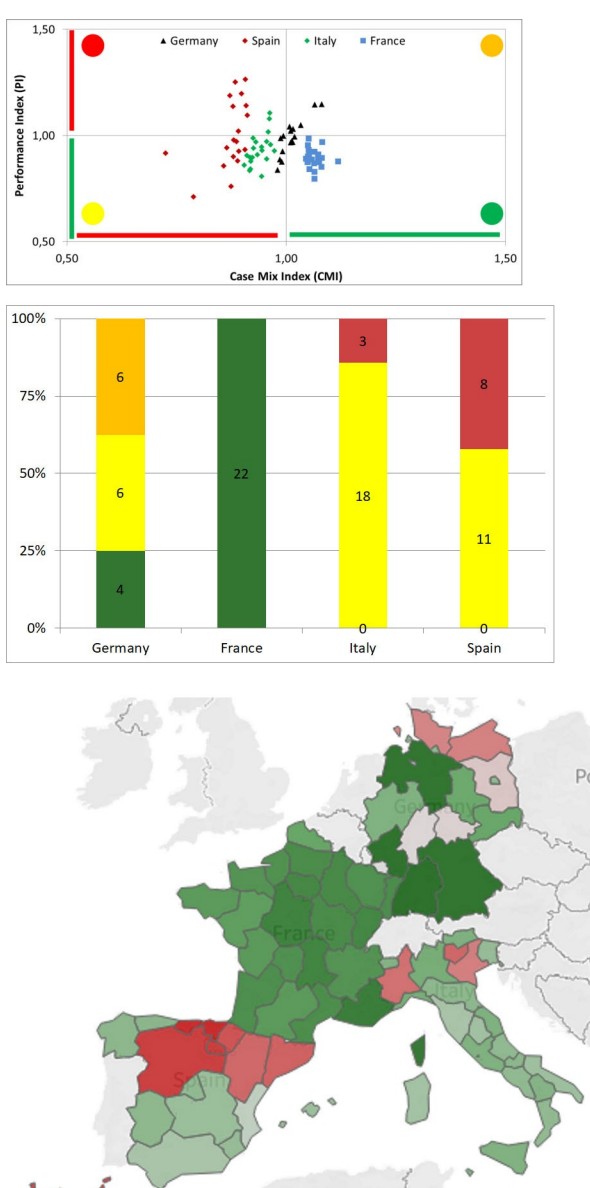

**Fig 6. Classification of countries' regions showing: (a) Case Mix and Performance indices diagram highlighting the four areas; (b) distribution of regions by country and by area and (c) map of the regions.** Each region is coloured on the basis of the ratio PI / CMI: dark red reported the lowest value, dark green reported the highest value (map was created on the basis of Eurostat data [23] using Tableau Software; Seattle, Washington, United States).

as well as of the complexity and performance (y-axis) analyses to capture the overall behaviour of each region in the management of hospital beds. Summarizing, the following overall findings can be reported:

- France: both complexity and performance indices are positively linked in all regions, while only a limited number of regions (mainly in the north) have an efficient management of beds in terms of turnover and occupancy. This low efficient hospital bed management can be due to the high number of beds and/or the limited number of hospitalizations that also limit the occupancy rate and the turnover interval.

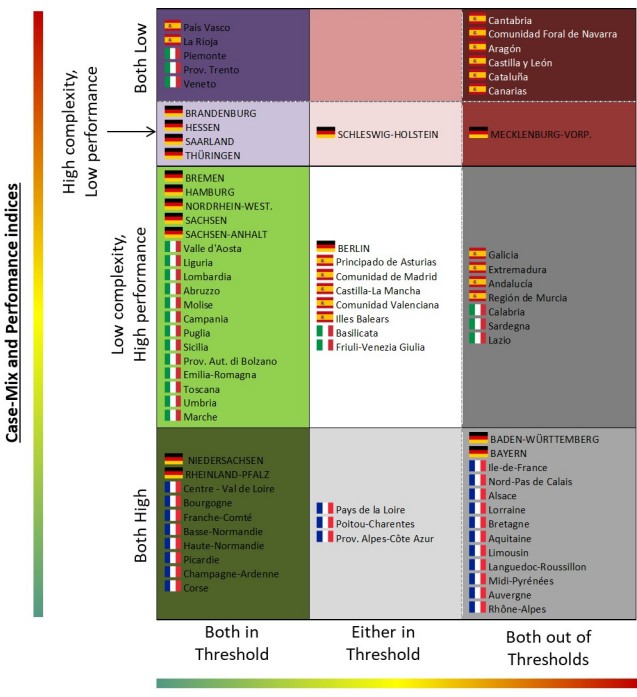

**Fig 7. Classification of each region within the matrix reported in Fig 1.**

- Germany: scattered distribution with regional differences across the country: the North where complex cases are not efficiently managed, while the South has low rates in high performance and in managing complex cases. In the centre of the country hospitals generally treat complex cases with low performance outcome resulting in an efficient management of hospital beds.

- Spain: similar results for all regions considering the hospital bed management and the complexity of cases. Southern regions plus 2 northern regions have a high performance. Regions with low complexity cases managed with high performance result in a low occupancy rate and turnover interval considering the low number of day that each patient spends in the hospital. On the contrary, there are regions where both hospital bed management and performance are below the benchmarking threshold, in particular when low complex cases are managed. This is mainly due to the high number of beds and/or limited number of hospitalizations that limit the occupancy rate and the turnover interval.

- Italy: the majority of regions have an efficient management of beds with some exceptions located in the north of the country. Considering the complexity of cases, the Italian regions tend to manage low complex hospitalization compared to the other European countries. However, they are efficiently managed in terms of performance almost in all Italian regions.

## Limitation of the study

This analysis presents limitations both from a methodological and data perspective. Firstly, the analysis of the complexity of cases treated at a regional level is based on the Case-Mix Index (CMI) that is computed taking into account the average length of stay of the different diseases addressed by the relevant structures. This approach can effectively reflect the patients' severity

of disease and difficulty of treatment in hospitals [36, 37]. However, these indexes focus the attention on the relative costs and resources needed to treat patients to support decision makers in the allocation of hospital resources, such as beds, personnel, devices [21]. Despite these premises the adoption of the CMI and PI can be considered a proxy to determine for each diagnosis its level of complexity as well as the potential risks and resources consumed at a regional level. The second limitation entails the availability of recently updated data considering both the structural components, which are generally not reported at a regional level and the hospitalization process that does not consider the continuous change in terms of bed availability across the European regions. This makes it difficult to apply this methodology in this pandemic period, whose analysis should be based on robust comparable data, taking into account the available resources in terms of equipment and specialized staff along with the whole health system's capacity of quickly rearranging emergency and ordinary treatments.

## Discussion and conclusions

The rapid global spread of the COVID-19 in the first months of 2020 has overwhelmed hospitals and local communities in Europe as well as worldwide. One of the main challenges was to rapidly and efficiently manage the health resources (i.e. medical professionals, equipment, hospital beds, general practitioners, etc.) to prevent overload and saturation. This is particularly important in countries that suffered from low number of available beds (saturated in the first days of the epidemics), shortage of health professionals (faced a high workload, job stress, time pressure, etc.) and limited organizational support [38, 39]. For these reasons, the analysis of hospital capacity and the efficiency in the management of its structural components before the emergency outbreak provides an important reference point to further explore how the management of the emergency has been carried out [8]. The analysis presented in this paper may be also used as a baseline providing indications on the best way to achieve hospital disaster preparedness in case of other possible future pandemics as it can highlight structural bottle necks and/or facilitators of national health systems [40]. The lack of hospital beds in different European countries has been attributed to a set of important cuts in the financing of public health over the last years with reduction of healthcare personnel both in the community and in the hospital care. Furthermore, the study provides an overview of the efficiency in the management of hospital beds across the regions in four European countries along with data highlighting differences between regions and countries in terms of structural health resources.

A first snapshot is provided through the study of the structural components available in each country also considering differences across years. The results of this analysis firstly show a clear difference between the northern countries (i.e. Germany and France) and the southern countries (i.e. Spain and Italy) in terms of both number of beds and hospital personnel per population. This gap is strictly related with the expenditure on health, particularly with the inpatient care. In general, all countries had a reduction of beds between 2017 and 2010: Spain was the country with the lowest reduction of beds (-1%) in comparison with Germany (-4%), France (-6%) and Italy (-7%). Regional differences have been detected: Sardegna in Italy was the region with the highest reduction (-13%), while La Rioja in Spain was the region with the highest increase (+13%). This trend may be explained, for instance, by two main organizational factors that have reduced the provision of hospital services: different scheduled procedures are mainly provided on a day hospital basis reducing the number of beds needed to treat the patients and the provision of health services is increasingly shifting from formal institutional facilities (e.g. hospitals) to home and territorial care. This is particularly evident in Italy and France that increased their healthcare expenditure on home and outpatient care. However, differences detected at regional level within the analysed countries may indicate a not

homogeneous coverage of health resources causing inequality. This may bias service provision, especially in critical periods and requires further analysis also in view of possible reorganization of resource distribution in periods of crisis. The same applies to the health professional decrease. In fact, if changes of service organization may compensate the reduction of bed availability, the lack of health personnel can have more serious effects undermining efforts in reallocation of resources during emergency. However, to determine whether and to what extent these aspects have influenced the health systems' response to the COVID-19 pandemics, more robust and time-series data are needed as well as a better knowledge on the virus physiopathology and treatment. Under this perspective, the analysis of hospital bed management proposed here provides an ex-ante analysis for future investigations on health system capacity to effectively reacts to emergency periods. Additional studies are needed relying on updated and robust data to provide indications on the hospital-territory relationship as well as to investigate whether regional differences in the efficiency in the management of hospital beds may have had an impact in the diffusion and mortality of COVID-19.

The study put in evidence that, in routinely periods, hospital beds are generally efficiently managed in particular in Italy and Germany where only a limited number of regions show a slow turnover and/or a low bed occupancy rate. This highlights that cases are handled with rapid shifts and without leaving beds empty during the year. From a territorial perspective, in Italy the majority of the efficient regions are in the northern part of the country, in Germany this is the case in the central part of the country. Differently, in France and Spain the majority of the regions display the turnover and/or the bed occupancy rate indices outside the relevant thresholds. In particular, in Spain only three regions are within the target, while in France beds are efficiently managed only in a limited number of northern regions.

Considering the complexity of cases and the performance in their management, in France all the regions manage complex cases with a high performance. Four regions in Germany provide this result, while six regions efficiently manage low complex cases. On the contrary, different regions in Italy and Spain show low performance rate even if managing low complex cases.

These considerations highlight that northern countries (i.e. France and Germany) could rely on the robust structural components of the hospital system during the epidemic outbreak, compared to southern countries (i.e. Italy and Spain). The number of hospital beds, staff in health centres and the greater investments in the health system may have positively impacted on the management of the COVID-19 diffusion. In particular, the high availability of beds in the majority of the France regions with a low occupancy rate and high turnover interval led these regions to have a high number of available beds across the country also considering that this country generally manages complex cases. German regions present a similar structural component as the number of available beds is significantly higher than in the other countries. However, differently from France, this country was less affected by the virus in terms of both number of cases and deaths until the 14th of July 2020. Moreover, despite the high availability of beds the hospitalization rate in the German regions was considerably lower than in the other countries. This approach may have helped this country in efficiently managing patients affected by the COVID-19 without exposing hospital wards and intensive care unit to a risk of saturation. The impact of the COVID-19 was completely different in Italy and Spain where a consistent number of patients were treated during the outbreak. From a structural perspective the regions of these two countries had to deal with a relevant large number of patients relying on a reduced number of both hospital beds and professionals. Moreover, a further critical factor for these south nations concerns the dissimilar distribution of cases across regions. In Italy the COVID-19 was concentrated in particular in the northern part of the country where also the majority of patients were hospitalized [8]. A similar distribution was detected also in Spain where the virus was concentrated in some regions (i.e. Madrid, Castilla) than in others (i.e.

Galicia, Canary Island). Even if in these countries the hospital beds were efficiently managed, the concentration of hospitalized patients in particular in the ICU and the scarcity of beds have put pressure on the hospital systems in particular in the regions most affected by the COVID-19.

## Supporting information

**S1 Table. Cross-regional comparison of the results of bed management as well as complex and performance analysis.** CMI = Case-Mix Index, PI = Performance Index, TOI = Turn-Over Interval, BOR = Bed Occupancy Rate, BTO = Bed Turn-Over, AvLOS = Average Length of Stay. Green cells specifies values that are within the relevant threshold: CMI > 1, PI < 1, 3<TOI<1, 75%<BOR<85%.
(DOCX)

## Author Contributions

**Conceptualization:** Fabrizio Pecoraro, Daniela Luzi, Fabrizio Clemente.

**Data curation:** Fabrizio Pecoraro.

**Formal analysis:** Fabrizio Pecoraro.

**Methodology:** Fabrizio Pecoraro.

**Supervision:** Fabrizio Pecoraro, Daniela Luzi, Fabrizio Clemente.

**Validation:** Fabrizio Pecoraro, Daniela Luzi, Fabrizio Clemente.

**Writing – original draft:** Fabrizio Pecoraro.

**Writing – review & editing:** Fabrizio Pecoraro, Daniela Luzi, Fabrizio Clemente.

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
