## [Decision Letter · Decision Letter 0]

21 Dec 2020

PONE-D-20-34140

The efficiency in the ordinary hospital bed management: A comparative analysis in four European countries before the COVID-19 outbreak

PLOS ONE

Dear Dr. Pecoraro,

Thank you for submitting your manuscript to PLOS ONE. After careful consideration, we feel that it has merit but does not fully meet PLOS ONE’s publication criteria as it currently stands. Therefore, we invite you to submit a revised version of the manuscript that addresses the points raised during the review process.

Several reviewers were approached. Regretfully, onlly one reviewer agreed and his overall impression is included below.  I took upon myself to act as a second reviewer and I do have some issues that need to be dealt with before accepting this manuscript for publication:

1. The manuscript needs some language revisions.  Sometimes the journal language editor may suggest the needed changes.  However, I suggest you find someone proficient in the English language who understands the subject, who will not change the meaning of the sentence. I will give you one example from the discussion.  You wrote: 

At the moment (i.e. October 8th, 2020), in Europe the virus, after a relative deceleration during the summer period, has continued to rise reaching 5.644.987 (15% over the world cases) confirmed cases and causing 228.053 (21%) deaths [1,4].

What does it mean 15% over the world cases.

2. The heading for figure 5 indicates taht 5b is a map while it is a histogram.  In 5c there is something the authors wanted to show at the right side - but what is it?

3. The heading for figure 6 indicates there are three components to the figure, while only one is presented.

Please review if your headings are correct.

We look forward to receiving your revised manuscript.

Kind regards,

Itamar Ashkenazi

Academic Editor

PLOS ONE

Journal Requirements:

3. We note that Figures 3a, 4 and 5a in your submission contain map images which may be copyrighted.

a. You may seek permission from the original copyright holder of Figures 3a, 4 and 5a to publish the content specifically under the CC BY 4.0 license. 

4. We note you have included a table to which you do not refer in the text of your manuscript. Please ensure that you refer to Table 1 in your text; if accepted, production will need this reference to link the reader to the Table.

Reviewers' comments:

Reviewer's Responses to Questions

**Comments to the Author**

1. Is the manuscript technically sound, and do the data support the conclusions?

Reviewer #1: Yes

2. Has the statistical analysis been performed appropriately and rigorously? 

Reviewer #1: Yes

3. Have the authors made all data underlying the findings in their manuscript fully available?

Reviewer #1: Yes

4. Is the manuscript presented in an intelligible fashion and written in standard English?

Reviewer #1: Yes

5. Review Comments to the Author

Reviewer #1: Its an interesting article and well written. It is unique in that it looks at the COVID-19 response from a perspective of already existing health systems capacity as it pertains to bed availability and bed management. And it gives an important perspective on health systems resillience.

6. PLOS authors have the option to publish the peer review history of their article (what does this mean?). If published, this will include your full peer review and any attached files.

Reviewer #1: **Yes: **Samuel Kidane,MD

---

## [Author Response · Author response to Decision Letter 0]

12 Feb 2021

Dear Editor,

We thank you for your email regarding our submission to the PLOS one journal. We are grateful for your suggestions and the comments of the peer reviewers.

We have answered the comments raised in a point-by-point format addressing any concerns. Comments are reported in the following. The manuscript has considerably improved by these modifications and we thank the reviewers for their attention. We have highlighted the changes in the main text.

Yours Sincerely,

Fabrizio Pecoraro

Reviewer 1

1. The manuscript needs some language revisions. Sometimes the journal language editor may suggest the needed changes. However, I suggest you find someone proficient in the English language who understands the subject, who will not change the meaning of the sentence. I will give you one example from the discussion. You wrote: 

At the moment (i.e. October 8th, 2020), in Europe the virus, after a relative deceleration during the summer period, has continued to rise reaching 5.644.987 (15% over the world cases) confirmed cases and causing 228.053 (21%) deaths [1,4]. What does it mean 15% over the world cases.

Author's response: In this specific case it means that Europe counts the 15% of all cases worldwide. Moreover, all the authors carefully read and improved the manuscript which was also revised by a professional mother tongue translator appointed by IRPPS.

2. The heading for figure 5 indicates that 5b is a map while it is a histogram. In 5c there is something the authors wanted to show at the right side - but what is it?

Author's response: a. The caption was not clear and it has been updated as follows: Figure 5. Efficiency results of countries’ regions in the management of hospital beds: (a) Barber-Johnson diagram highlighting the three areas and the centroid (TOI = 2 and BOR = 80%); (b) distribution of regions by country and by area and (c) map of the regions. Each region is coloured on the basis of the Euclidean distance from the centroid: dark red reported the highest distance, dark green reported the lowest distance (map was created using Tableau Software; Seattle, Washington, United States). 

3. The heading for figure 6 indicates there are three components to the figure, while only one is presented. Please review if your headings are correct.

Author's response: a. Figure 6 were wrongly sent. The one reported in the submission was Figure 7 instead. Moreover, the caption was not clear and it has been updated as follows: Figure 6. Classification of countries’ regions showing: (a) Case Mix and Performance indices diagram highlighting the four areas; (b) distribution of regions by country and by area and (c) map of the regions. Each region is coloured on the basis of the ratio PI / CMI: dark red reported the lowest value, dark green reported the highest value (map was created using Tableau Software; Seattle, Washington, United States). 

Journal Requirements:

Author's response: Manuscript has been updated on the basis of the above mentioned requirements 

Author's response: Data are already organized in an Excel spreadsheet which has been already uploaded on Zotero database (doi: 10.5281/zenodo.3737839) and is ready to be published. Data availability statement will be updated as soon as the manuscript would be accepted for publication. 

3. We note that Figures 3a, 4 and 5a in your submission contain map images which may be copyrighted. 

Author's response: Maps were created using Tableau Software; Seattle, Washington, United States. This information has been added to the caption of the relevant figures. 

4. We note you have included a table to which you do not refer in the text of your manuscript. Please ensure that you refer to Table 1 in your text; if accepted, production will need this reference to link the reader to the Table.

Author's response: Text updated.

---

## [Editor Report · Decision Letter 1]

8 Mar 2021

The efficiency in the ordinary hospital bed management: A comparative analysis in four European countries before the COVID-19 outbreak

PONE-D-20-34140R1

Dear Dr. Pecoraro,

We’re pleased to inform you that your manuscript has been judged scientifically suitable for publication and will be formally accepted for publication once it meets all outstanding technical requirements.

Kind regards,

Itamar Ashkenazi

Academic Editor

PLOS ONE
---

## [Editor Report · Acceptance letter]

12 Mar 2021

PONE-D-20-34140R1 

The efficiency in the ordinary hospital bed management: A comparative analysis in four European countries before the COVID-19 outbreak 

Dear Dr. Pecoraro:

I'm pleased to inform you that your manuscript has been deemed suitable for publication in PLOS ONE. Congratulations! Your manuscript is now with our production department. 

Kind regards, 

on behalf of

Dr. Itamar Ashkenazi 

Academic Editor

PLOS ONE